



**A Fast-response automated gas equilibrator (FaRAGE) for continuous *in situ***
**measurement of methane dissolved in water**
Shangbin Xiao[1], Liu Liu[2]*, Wei Wang[1], Andreas Lorke[1,3], Jason Woodhouse[2] and Hans-Peter
Grossart[2]*
*[1] College of Hydraulic & Environmental Engineering, China Three Gorges University, 443002 Yichang, China*
*[2] Department of Experimental Limnology, Leibniz Institute of Freshwater Ecology and Inland Fisheries, 16775*
*Stechlin, Germany*
*[3] Institute for Environmental Sciences, University of Koblenz-Landau, 76829 Landau, Germany*
*Corresponding authors
Emails: liu.liu@igb-berlin.de; grossart@igb-berlin.de
**Abstract**

12        Biogenic methane ($CH_4$) emissions from inland waters contribute substantially to

global warming. In aquatic systems, $CH_4$ dissolved in freshwater lakes and reservoirs is
highly heterogeneous both in space and time. To better understand the biological and physical
processes that affect sources and sinks of $CH_4$ in lakes and reservoirs, dissolved $CH_4$ needs to
be measured with a highest temporal resolution. To achieve this goal, we developed the **Fast-**
**Response Automated Gas Equilibrator (FaRAGE)** for real-time *in situ* measurement of
dissolved $CH_4$ concentration at the water surface and in the water column. FaRAGE can
achieve an exceptionally short response time ($t_{95\%}$ = 12 s when including the response time of
the gas analyzer) while retaining an equilibration ratio of 63% and a measurement accuracy of
0.5%. An equilibration ratio as high as 91.8% can be reached at the cost of a slightly
increased response time (16 s). The FaRAGE is capable of continuously measuring dissolved
$CH_4$ concentrations in the nM-to-mM ($10^{-9}$ - $10^{-3}$ mol $L^{-1}$) range with a detection limit of sub-



nM ($10^{-10}$ mol L$^{-1}$), when coupled with a cavity ring-down greenhouse gas analyzer (Picarro
GasScouter). It enables the possibility of mapping dissolved $CH_4$ concentration in a "quasi"
three-dimensional manner in lakes. The FaRAGE is simple to operate, inexpensive, and
suitable for continuous monitoring with a strong tolerance to suspended particles. The easy
adaptability to other gas analyzers such as Ultra-portable Los Gatos and stable isotopic gas
analyzer (Picarro G2132-i) also provides the potential for many further applications, e.g.
measuring dissolved $^{13}\delta C\text{-}CH_4$ and $CO_2$.



## 1 Introduction

Despite the well-established perception of inland waters as a substantial source of atmospheric methane ($CH_4$) (Bastviken et al., 2011; Cole et al., 2007; Tranvik et al., 2009), large uncertainties remain owing to poorly constrained sources and sinks (Saunois et al., 2019). Most freshwater lakes and reservoirs are often oversaturated with $CH_4$ (relative to atmosphere) and its distribution is characterized by high spatio-temporal heterogeneities (Hofmann, 2013). Point-based and short-term measurements can result in biases in estimating diffusive $CH_4$ flux (Paranaíba et al., 2018). Thus, resolving the spatio-temporal dynamics of dissolved $CH_4$ concentration in lake water is a prerequisite for better budgeting sources and sinks in freshwater lakes.

Methane within lakes is often characterized by pronounced vertical and horizontal concentration gradients, which can occur either below or above thermocline. In many deep stratified lakes, a sharp vertical gradient below the thermocline can develop in the anoxic hypolimnion (mM range) (Encinas Fernández et al., 2014; Liu et al., 1996). In contrast, in some stratified lakes with a fully oxygenated hypolimnion $CH_4$ can accumulate above the thermocline (~µM range) (Grossart et al., 2011; Donis et al., 2017; Günthel et al., 2019). The concentration of dissolved $CH_4$ is also regulated by loss due to oxidation and emission to the atmosphere (Bastviken et al., 2004; Juutinen et al., 2009). Both rates can be highly variable, particularly for the flux term which is strongly affected by wind and convective mixing (Read et al., 2012; Vachon and Prairie, 2013). In addition to the uneven vertical $CH_4$ distribution, apparent horizontal gradients have been observed in lakes where littoral sediments are identified as a $CH_4$ source (Murase et al., 2003). This horizontal $CH_4$ gradient can also contribute to the epilimnetic $CH_4$ peak in pelagic waters via lateral transport (Hofmann et al., 2010; Fernández et al., 2016; Murase et al., 2005; Peeters et al., 2019). Nevertheless, dissolved $CH_4$ in lake water is not only featured with variable spatial patterns, it also changes





at different time scales as most processes that contribute to the spatial heterogeneity are not
always synchronized.
The rise and fall of lake $CH_4$ concentration often show strong seasonality that are
driven primarily by thermal stratification (Encinas Fernández et al., 2014) and phytoplankton
dynamics (Günthel et al., 2019). While the build-up of hypolimnetic $CH_4$ storage is a slow
process that is closely related to the development of lake hypoxia, the epilimnetic $CH_4$
maximum can be highly variable even at a daily basis as it is strongly affected by
phytoplankton dynamics (Günthel et al., 2019; Hartmann et al., 2020; Bižić et al., 2020). In
addition, storms can act as another driver for short-term $CH_4$ dynamics in the lake because it
often leads to higher evasion rates caused by strong vertical turbulent mixing (Zimmermann et
al., 2019) and enhanced horizontal transport (Fernández et al., 2016). While the seasonal
patterns of dissolved $CH_4$ concentration in lake water seem recurrent and can be simulated
(Bartosiewicz et al., 2019), the unpredictable effects of short-term phytoplankton dynamics
and storm events can present a challenge in modeling lake $CH_4$ dynamics.
While there is urgent need for resolving the spatio-temporal variabilities of $CH_4$ in
large water bodies (e.g. deep, stratified lakes), we recognize limitations in the available
methodology. Like most gases in dissolved phase, $CH_4$ cannot be measured directly in water.
Instead, a carrier gas (synthetic air or at air concentration) is added to achieve (full/partial)
gas-water equilibration. The headspace gas sample is then measured with a gas spectrometer
and the concentration of targeted gas can be calculated according to Henry's law (Magen et
al., 2014). To save sampling effort, continuous gas equilibration devices have been developed,
which generally can be classified to four categories: 1) Membrane type (Schlüter and Gentz,
2008; Boulart et al., 2010; Gonzalez-Valencia et al., 2014; Hartmann et al., 2018) - gases are
extracted from water by using a gas-permeable membrane; 2) Marble type (Frankignoulle et
al., 2001; Santos et al., 2012) - gas exchange is enhanced by pumping water through marbles

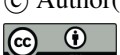



that increases the gas-water contact area; 3) Bubble type (Schneider et al., 1992; Körtzinger et
al., 1996; Gülzow et al., 2011) - dissolved gases are stripped out by bubbling the water sample;
4) Showerhead type (Weiss-type) (Johnson, 1999; Rhee et al., 2009; Li et al., 2015) - water is
pumped from top and then mixed with a circulated headspace carrier gas. A full evaluation on
the performance of these devices was provided in a recent review (Webb et al., 2016), where,
the most important paprameter, response time, was found to vary between 2-34 min for
dissolved $CH_4$. While it is already encouraging, improvements are expected to further shorten
the response time.
Driven by the need to resolve temporal and spatial variabilities of dissolved $CH_4$ in
lakes/reservoirs with sufficient precision, we developed a novel, low-cost equilibrator to
achieve fast gas-water equilibration. The **Fast-Response Automated Gas Equilibrator**
**(FaRAGE)** can be coupled with a portable gas analyzer, which makes it perfect for field use.
Here, the performance of the FaRAGE is evaluated by investigating its response time,
detection limit and equilibration ratio. Applications are provided exemplarily to demonstrate
the potential of the FaRAGE for improving our understanding on the spatial distribution and
temporal dynamics of dissolved $CH_4$ in inland waters.
**2 Materials and Methods**
**2.1 Device description**
The design of the FaRAGE is modified from two types of equilibrators: Bubble-type
(Schneider et al., 1992) and Weiss-type (Johnson, 1999). In contrast to the traditional bubble-
type and Weiss-type equilibrators that create a large-volume headspace and circulates air back
to the headspace, the FaRAGE is a flow-through system that adds gas flow into a constant
water flow to produce a minimal headspace for continuous concentration measurement of
$CH_4$ dissolved in water.





The operation principle of the FaRAGE is depicted in Fig. 1 and photos of the main
parts of the prototype are provided in Fig. S1. A list of information on suppliers and cost of
each part can be found in Table S1. A mass flow controller (SIERRA C50L, Netherlands) is
used to generate a constant carrier gas (normal air/synthetic air) flow (1 L min$^{-1}$) from a
compressed air tank coupled with a pressure regulator. Water samples are taken continuously
using a peristaltic pump (500 mL min$^{-1}$), and the flow is monitored using a flow meter
(Brooks Instrument, Germany). The two flows mix in a gas-water mixing unit that is
composed of a gas bubble generating unit and a coiled hose for further gas-water turbulent
mixing. In the bubble unit (modified from a 10 mL plastic syringe), a jet flow is created by
adapting narrowed tubing (2 mm inner diameter) to the water pumping hose (3.2 mm inner
diameter). Degassing occurs when the jet flow enters the chamber with a sudden enlarged
diameter (14 mm). Degassing is further enhanced by micro-bubbles that are generated by a
bubble diffusor attached to the carrier gas hose (inside the bubble unit). The gas-water
mixture flows through the 2-m long Tygon tube (3.2 mm inner diameter) where additional
equilibration occurs. The flow is finally introduced to a gas-water separation unit (a 30 mL
plastic syringe) where the headspace gas is separated from the water. In this chamber, water
falls down freely to the bottom while the headspace gas is taken directly to a greenhouse gas
analyzer (1 L min$^{-1}$ gas pumping rate; GasScouter G4301, Picarro, USA). A 2-m long Tygon
tube (3.2 mm inner diameter) is attached to the top of the chamber for venting excess gas flow
while stabilizing gas pressure in the headspace. The bottom water is discharged back to the
lake using another peristaltic pump (500 mL min$^{-1}$). To protect the gas analyzer from
damaging high water vapor content, a Teflon membrane filter (pore size 0.2 μm) is placed
before the gas intake (resulting in a ~210 mL min$^{-1}$ reduction in flow rate of gas sample,
which is vented from the bypass at the top of the gas separation unit). A desiccant (a 20 mL
plastic syringe filled with dried silicone beads) is used to reduce moisture concentration to <
0.1% when attaching to a Picarro G2132-i isotope analyzer (Picarro, USA), in which < 1%





moisture level is required for $^{13}\delta$C-CH$_4$ measurement. The temperature of the water sample at
the point of equilibration with the headspace gas is monitored using a fast thermometer
(precision 0.001 °C, 1 Hz, TR-1050, RBR, Canada) attached to the end of water discharging
hose.

As concerns might arise from the availability of gas analyzer coupled to the FaRAGE,

in addition to Gas Scouter from Picarro, two additional widely used models of greenhouse gas
analyzers were tested. They are the Ultraportable Los Gatos (Los Gatos Research, USA) and
stable isotopic CH$_4$ analyzer (G2132-i, Picarro, USA). The main technical details of all three
tested gas analyzers are listed in Table S2.

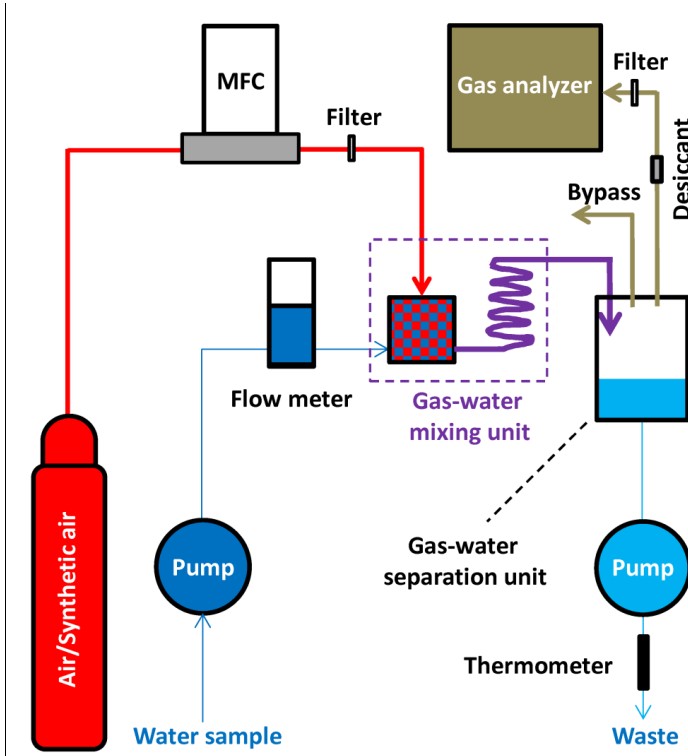


**Fig. 1** Schematic design of the FaRAGE. The components include: Air tank containing
compressed carrier gas (air or synthetic air) with a pressure regulator, a mass flow controller





(MFC) for generating constant carrier gas flow, two peristaltic pumps for taking and
discharging water, respectively, a flow meter for monitoring water sample flow, a gas-water
mixing unit, a gas-water separation unit, a gas analyzer, and a thermometer for measuring
water temperature at phase equilibration. A Teflon membrane filter is placed after the MFC
and another is added before the gas analyzer to protect from being flooded. A desiccant is
used to dry the gas flow to the gas analyzer (if Picarro isotopic analyzer is used). The red
color marks the flow of carrier gas, dark blue line indicates the water sample, purple line
shows the flow of gas-water mixture, the light brown line shows the flow of gas sample (after
partial equilibration) and the light blue line depicts the water discharged back to lake. The
thickness of the lines scales with the gas/water flow rates. The arrows show the flow
directions.
**2.2 Lab validation**
The FaRAGE prototype was first tested intensively in the lab to determine both the
equilibration ratio and response time. The equilibration ratio is defined as the percentage of
the gaseous $CH_4$ concentrations at the outlet of the gas equilibrator in comparison to the
equilibrium concentration (full gas-water equilibration). The equilibration ratio was
established by measuring a range of $CH_4$ stock solutions (nano-to-micro molar dissolved gas
concentrations). These standard solutions were prepared by adding different amounts of $CH_4$
into a 200 mL headspace of a 2 L Schott bottle filled with Milli-Q water. The exact dissolved
$CH_4$ concentrations in these solutions were tested with the traditional manual headspace
method: a 400 mL headspace was created in a 500 mL plastic syringe with nitrogen gas. The
$CH_4$ concentration of the headspace gas was then measured using GasScouter G4301 (Picarro,
USA). At the same time, $CH_4$ concentrations of these standard solutions were measured with
the FaRAGE for at least 2 min and an average was calculated from more than 60 individual



data points. For simplicity, we directly compared dissolved $CH_4$ concentrations measured
using the two different methods, i.e., our equilibrator and manual headspace method.
The response time of the device was investigated by switching the water sample inlet
between two water samples with different $CH_4$ concentrations. Triplicated measurements
were performed. An exponential fit was applied to the concentration change curve and the
response time was determined as time needed to reach 95% of the final concentration.
The effect of water-to-gas mixing ratio on equilibration ratio and response time of the
device was investigated. By fixing the carrier gas flow rate to 1 L min$^{-1}$, the water-to-gas
mixing ratio was varied from 0.04, 0.08, 0.12, 0.15, 0.24, 0.29, 0.36, 0.43 and 0.5 by adjusting
the water sample flow rate. The effect of tube length on performance of the device was also
examined by adapting 1, 2, 4.4 and 8.4 m Tygon tube onto the gas-water mixing unit. For all
these tests, triplicated measurements of the equilibration ratio and response time were
performed corresponding to different mixing ratios and the mean values were used for
analysis.
Tests were performed to investigate the performance of the device when adapting to
two other types of gas analyzers. As the equilibration ratio is unaffected by the model of gas
analyzers, only response time was determined. This was done by fixing carrier gas and water
sample flow rates to 1 and 0.5 L min$^{-1}$, respectively. The surplus gas was vented to the air as
Ultraportable Los Gatos and Picarro G2132-i have a gas intake flow rate of only 500 and 25
mL min$^{-1}$, respectively. The effect of desiccant on response time of Picarro G2132-i was
checked by measuring gas samples with and without a desiccant installed.
**2.3 Field tests**
Two lakes in Germany were chosen for field test. Lake Stechlin is a deep meso-
oligotrophic lake with a maximum depth of 68 m and Lake Arend is a eutrophic lake with a





maximum depth of 48 m. Pronounced $CH_4$ peaks in the epilimnion of Lake Stechlin have
been previously reported that were measured with various methods (Grossart et al., 2011;
Hartmann et al., 2018; Tang et al., 2014). This makes it ideal for our testing purpose. While
$CH_4$ profiles at Lake Arend have never been reported, the metalimnetic oxygen minimum in
the lake observed during summer (Kreling et al., 2017) renders it interesting for $CH_4$ profiling
throughout the entire water column.

Due to the high potential of the FaRAGE for real-time *in situ* measurement of

dissolved $CH_4$ concentrations, we explored potential field applications. These field tests
included depth profiling of dissolved $CH_4$ concentrations in Lake Arend and Lake Stechlin
and investigations of the horizontal distribution of surface dissolved $CH_4$ concentration across
the entire Lake Stechlin. For the first application, a fast-response CTD (conductivity,
temperature and depth) profiler (XR-620 CTD+, RBR, Canada) was mounted onto a winch
with a 30 m long water hose (4 mm inner diameter) attached. The CTD profiler with hose was
lowered down continuously at a constant speed (1 m min$^{-1}$). The exact depth and temperature
of sampled water can be extracted from the CTD profiler by correcting for the travel time of
water sample flow in hose. For the spatial mapping, a GPS antenna (Taoglas, AA.162, USA)
was attached to the Picarro gas analyzer. The water intake was submerged 0.5 m below the
water surface together with the CTD profiler and fixed to one side of the boat. The boat was
driven at a constant speed of 5 km h$^{-1}$.
**2.4 Theoretical background and data processing**

The FaRAGE shares a similar working principle to the Weiss-type gas equilibrator

described by Johnson (1999). The theoretical background and equations are provided in S3.

A simplified calculation is described by referring to the manual headspace method. In

principle the gas-water mixture is analogous to the static headspace method with the final $CH_4$





concentration in the gas phase assumed to reach a full equilibrium with that dissolved in the
aqueous phase. Therefore, by specifying the mixing ratio of air and water, the total mass of
$CH_4$ can be calculated by summing up the $CH_4$ in the headspace with the dissolved $CH_4$ (at
equilibrium according to Henry's law, which is temperature and pressure dependent) in the
aqueous phase and subtracting the mass of background $CH_4$ (from carrying gas with known
concentration). The dissolved $CH_4$ concentration is then expressed as the volumetric
concentration of total net mass of $CH_4$ in the dissolved phase in the given sample volume. A
separated exemplary calculation sheet (excel file S5) is provided, which allows for correction
for temperature and pressure change.
As the equilibration is only partially reached ($< 92\%$), a correction coefficient is
needed. This can be obtained by measuring the water samples with known concentrations
across a large gradient. By referring to the results measured with the manual headspace
method assuming full equilibration (Magen et al., 2014), an equation for precise correction of
the measured $CH_4$ concentrations can be obtained.
**3 Results and Discussion**
**3.1 Detection limit, equilibration ratio and response time**
The FaRAGE is capable of achieving a high gas equilibration ratio. We observed a
high correlation ($R^2 = 0.999$, $p < 0.01$) between the concentrations obtained using the
traditional headspace method and those measured using the FaRAGE (Fig. 2a) across a wide
range of dissolved $CH_4$ concentrations. The measurement accuracy is 0.5% (standard
deviation in relation to final concentration) once a stable plateau was reached (Fig. 2b). The
FaRAGE reaches a high equilibration ratio (63%) and ensures a rapid response. The
determined response time $t_{95\%}$ is only 12 s when switching from low-to-high (nano-to-micro
molar) dissolved $CH_4$ concentrations while the $t_{95\%}$ is a little longer ($15 \pm 2$ s) when switching



from high-to-low concentration (Fig. 2b). For the current design specifications that allow for a
high equilibration ratio, the detection is theoretically limited by the sensitivity of the coupled
gas analyzer. In the lab tests, a clear response was observed at least for $CH_4$ concentration at
air saturation (16.9 nM inside the lab building). The measureable $CH_4$ concentrations should
be at least sub-nM ($10^{-10}$ mol $L^{-1}$) given the high performance of cavity-ring-down gas
analyzers. This is more than sufficient for applications in inland waters where dissolved $CH_4$
concentrations are often above air saturation.

The response time for the FaRAGE results from two components: 1) the response of

the gas analyzer to changes in gas concentration and 2) the physical gas-water exchange
process. The response time for the gas analyzer is 5 s when the $CH_4$ concentration increases
(Fig. S2). The FaRAGE itself needs < 10 s to reach 95% of the final steady-state
concentration.

Equilibration ratio and response time of the FaRAGE is not sensitive to water-to-gas

mixing ratio (Fig. 2c) but rather to the length of the tube in the gas-water mixing unit (Fig. 2d).
Little effect was observed on the equilibration ratio in response to the increase of water-to-gas
mixing ratio. Also, the increase of water-to-gas mixing ratio did not significantly change the
response time of the device (on average 9 s for low-to-high and 13 s for high-to-low,
respectively). This is in contrast to other types of equilibrators in which an increase of water-
to-gas mixing ratio was found to result in a faster response (Webb et al., 2016). However, a
sharp enhancement of equilibration ratio was observed due to the extended length of the tube
for the gas-water mixing unit. A 91.8% equilibration ratio can be achieved by extending the
tube length to 13 m while extended response times (low-to-high 17 s and high-to-low 47.5 s,
respectively). Further enhancement of the equilibration ratio was not possible when a longer
tube (e.g. 18 m) was used. The gas flow rate cannot be stabilized at 1 L $min^{-1}$ due to the
increased resistance in response to the further extension of tube length.


As shown in Table S2 and Fig. S2, the fast response of the FaRAGE is partly due to
the extremely fast response of the Picarro Gas Scouter. This makes it unfair to compare with
other equilibrators in which different gas analyzers were used. Tests were performed by
adapting the FaRAGE to two other greenhouse gas analyzers (Ultraportable Los Gatos and
Picarro G2132-i) and the response times are listed in Table S3. Comparisons were made in
Webb et al. (2016) and Hartmann et al. (2018) where both $CH_4$ and $^{13}\delta C\text{-}CH_4$ were measured
using a Picarro G2201-i (Picarro, USA). Here we used a similar Picarro stable isotopic gas
analyzer (Picarro G2132-i) and unified all previous reported response time $\tau$ to $t_{95\%}$ by
applying the equation $t_{95\%} = 3\tau$. The comparison between up-to-date previous studies and this
study (Table S4) demonstrated the extraordinary fast response relative to all existing gas
equilibration devices. A 53 s response time was achieved when the FaRAGE was adapted to
the Picarro G2132-i, which is significantly faster than others (171-6744 s).

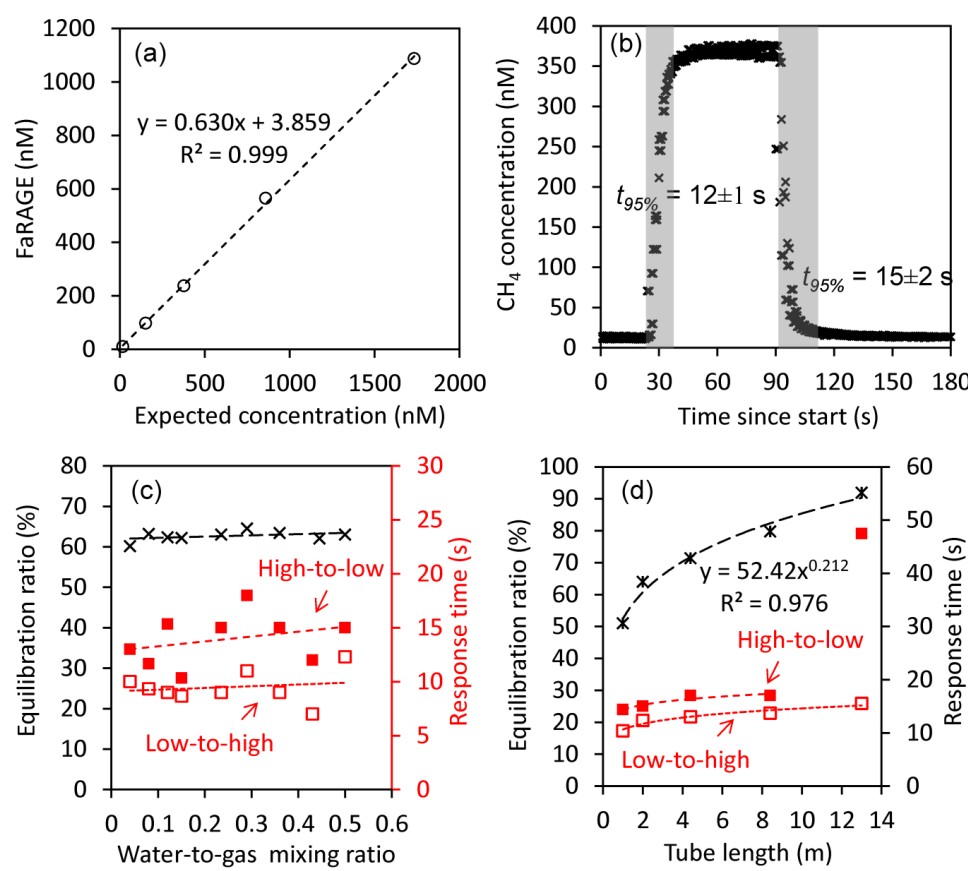


**Fig. 2** Performance of the Fast-Response Automated Gas Equilibrator (FaRAGE). (a) Exemplary correlation between measurements with the FaRAGE (with a 2-m tube in the gas-water mixing unit) and expected concentrations measured using the manual headspace method. (b) Exemplary response time of FaRAGE for low-to-high and high-to-low concentration changes (with a 2-m tube in the gas-water mixing unit; water-to-gas mixing ratio 0.5). Triplicated tests were performed and averaged response time was taken at the time point when 95% of the final concentration was reached. (c) Equilibration ratio and response time in response to changing water/gas mixing ratio (with a 2-m tube in the gas-water mixing unit). Black cross symbols are equilibration ratios, and low-to-high and high-to-low response times are represented by red open and solid squares, respectively. (d) Equilibration ratio and





response time in response to changing tube length of gas-water mixing unit (with a fixed
water-to-gas mixing ratio of 0.5). Black cross symbols are equilibration ratios, and low-to-
high and high-to-low response times are represented by red open and solid squares,
respectively.

### 3.2 Depth profiles of dissolved $CH_4$ from multiple lakes

Good agreement was observed between depths profiles of dissolved $CH_4$ concentration
measured using two different methods (Fig. 3). The observed occurrence of a maximum in the
vertical profile of dissolved $CH_4$ concentration in the upper layer of Lake Stechlin (Fig. 3b) is
consistent with previous observations (Grossart et al., 2011; Tang et al., 2014; Hartmann et al.,
2018). In Lake Arend we also observed a $CH_4$ peak (Fig. 3a), although the overall
concentration was lower. In contrast, with the headspace method the FaRAGE allowed for the
localized $CH_4$ concentration maximum to be described at a high vertical resolution, similar to
that obtained with more sophisticated membrane filter equilibrators (Hartmann et al., 2018;
Gonzalez-Valencia et al., 2014). The FaRAGE was capable of resolving differences in
dissolved $CH_4$ concentration in lake water at decimeter scales with ease. Whilst care should
be taken to ensure the sampling hose moves smoothly and slowly through the water column,
continuous profiling of a 20 m deep lake can be completed in 30 min. This is a big advantage
since *in situ* $CH_4$ concentrations can vary at very short time scales (hours to days) subject to
internal production, oxidation, weather conditions and etc. (cf. Hartmann et al. (2020)).




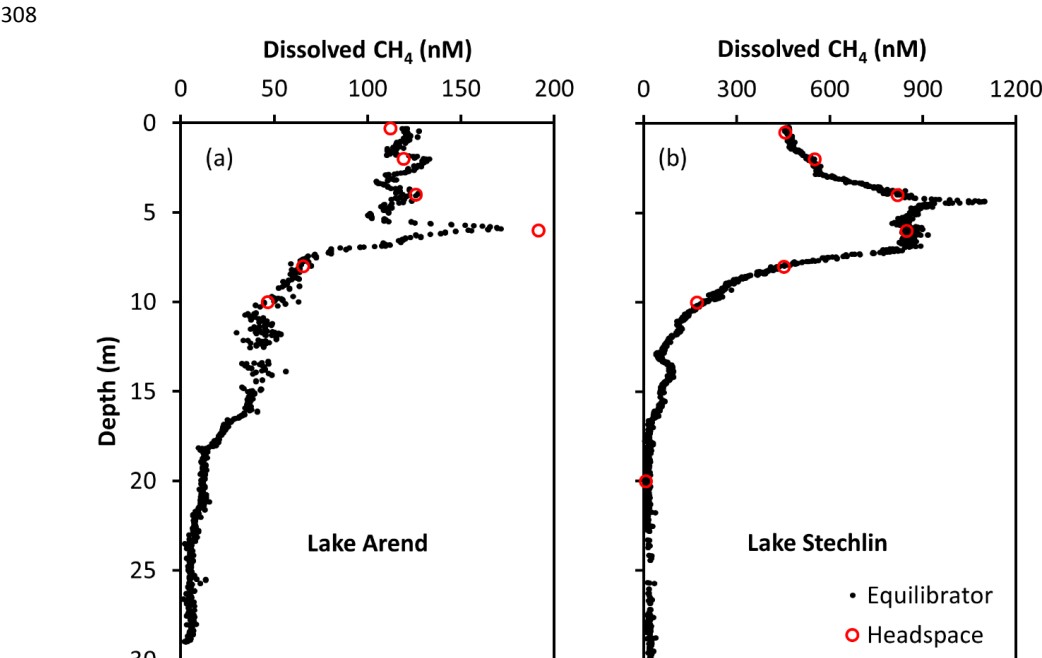

**Fig. 3** Depth profiles of dissolved $CH_4$ concentration from two lakes in Germany: (a)
eutrophic Lake Arend on June 17, 2019 and (b) meso-oligotrophic Lake Stechlin on July 23,
2019. Results from the headspace method are designated as red open circles and
measurements using the FaRAGE are shown as solid black dots.
**3.3 Resolving spatial variabilities of dissolved $CH_4$ concentrations**
We confirmed the capability of the FaRAGE to operate continuously over a 7-h period
without notable decreases in performance (Fig. 4a). Benefitting from its fast response rate,
surface water $CH_4$ concentrations across the 4.52 $km^2$ Lake Stechlin was mapped with great
detail within one day. During the cruise, 10 reference measurements were made at different
times, which were consistent with nonstop online *in situ* measurements. The cruising survey
demonstrated the capability of this device for resolving not just vertical dynamics of $CH_4$ in
lake water, but also the potential for studying horizontal distributions of $CH_4$ across large
distances, for instance large lakes and rivers. With a driving speed of 5 km h$^{-1}$ and a response
time of 12 s, a spatial resolution of 17 m can be achieved, which is sufficient for such a
medium-sized lake. The relative higher dissolved CH$_4$ concentrations in the shallow littoral
zone of Lake Stechlin (Fig. 4b) reflect higher CH$_4$ release from the local sediment.

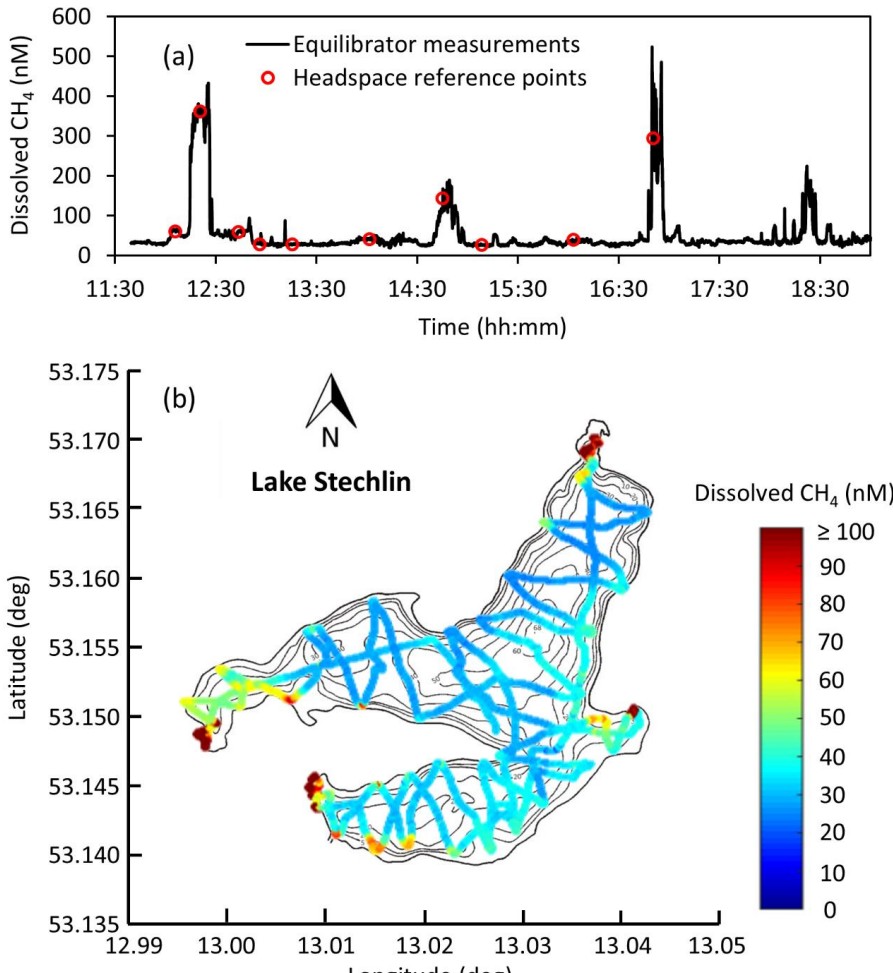


**Fig. 4** Map of surface dissolved CH$_4$ concentration at Lake Stechlin. (a) Time series of 7-h
continuous surface water CH$_4$ measurement on March 28, 2019. The reference headspace
measurements are shown as red circles. (b) Spatial distribution of surface water CH$_4$
concentration is given on top of the lake's bathymetry. Colored symbols show CH$_4$





concentration according to the color bar. Black lines show the outline of the lake with depth
contours.

**4 Comments and Recommendations**

**4.1 Adaptability to different gas analyzers**

The reasons for the significantly shortened response time of the FaRAGE compared to

other types of gas equilibrators are two-fold. While the working principle of the FaRAGE is
based on the bubble-type (Schneider et al., 1992) and Weiss-type equilibrators (Johnson,
1999), a reduced headspace volume is adopted, which enhances the physical gas-water
exchange. Another reason is the use of extremely fast-response gas analyzer (Picarro Gas
Scouter 4301). It is a highly recommended combination for concentration measurement when
the best time-wise performance is preferred due to its great mobility (Table S2). However,
coupling to other Cavity-Ring-Down gas analyzers is also possible (Table S3). This feature
enables a possibility to investigate stable isotopic nature of dissolved $CH_4$, which is important
when sources of $CH_4$ need to be identified.

When a portable gas analyzer (Picarro Gas Scouter or Ultraportable Los Gatos) is used

for measuring $CH_4$ concentration only, the gas equilibrator can be optimized for different
application environments. The length of coiled tube for gas-water mixing can be adjusted to
change the response time (Fig. 2d). For smaller lakes a higher spatial resolution can be
obtained by shortening the equilibration tubing, which shortens the response time, and hence
increases the spatial resolution, whilst maintaining an acceptable equilibration ratio (51%
when tube length is 1 m). In environments with extremely low dissolved $CH_4$ concentrations,
e.g. ocean waters, a longer gas-water mixing tube should be used to ensure a high gas
equilibration ratio.





To measure stable isotopic $CH_4$ in water, the sensitivity of the FaRAGE can be
modified to better adapt to the choice of gas analyzer (e.g., when Picarro G2201-i or G2132-i
is used). For example, high dissolved $CH_4$ concentrations (e.g. μM-to-mM range) can be
measured with greater accuracy by increasing the flow rate of the carrier gas relative to the
sample water flow, therefore diluting the $CH_4$ concentrations to the range of the gas analyzer.
This can be particularly useful, for instance, when an instrument has an optimal precision at a
low concentration range (1.8-12 ppm for Picarro isotopic gas analyzer) for $^{13}\delta C$-$CH_4$
measurements. By using pure $N_2$ gas or carrier gases (e.g. Helium and Argon) and
corresponding gas analyzers, it would be possible to measure other dissolved gas
concentrations, e.g. $CO_2$ can be measured simultaneously ($CO_2$ was tested in this study, but
not shown for simplicity). In addition, benefited from the high equilibration ratio of this
device (max. 91.8%), it would be possible to measure dissolved $CH_4$ (and other gases) close
to equilibrium concentrations.

**4.2 Uncertainties due to suspended solids, temperature and pressure change**

The FaRAGE is proven to be resistant to suspended solids in freshwater lakes without
having to use additional accessories. As shown in Fig. S3, apparent phytoplankton blooms
were observed in the two studied lakes each with a high biomass (Chl-a > 30 μg $L^{-1}$) in the
epilimnetic water. The measurements were unaffected, without any interruptions during
measurements. As algal particles are a large component in freshwater systems, it is safe to
claim the resistance of this device to suspended solids in such a system. However, care must
be taken to avoid the water intake hose hitting the bottom sediment, which could cause
blockage of the water hose.
The temperature and hydrostatic pressure could both change when water is pumped
out through a water hose. To consider the temperature effect, a fast temperature logger is used



(Fig. 1) which allows for corrections in calculation. Instead of using *in situ* lake temperature,
the temperature measured at the gas equilibrator should be used where gas equilibration
occurs. Our measurements found a minor effect when measuring surface waters but an
apparent warming for hypolimnetic water in deep lakes. While a calibration can be done
directly by taking water samples from multiple depths of the lake (e.g., Fig. 3) to consider this
effect, one could make the calculation without taking many samples by applying temperature
correction.

The temperature correction can be made by referring to the manual headspace method.

The constant gas and water flow can be used as headspace and water volume, respectively. By
considering the temperature and pressure effects on gas solubility, the dissolved $CH_4$
concentrations can be calculated (an example calculation sheet is provided in Table S5). The
calibration curve can be established using the manual headspace measurements as standards.
The final concentrations can be corrected for partial equilibration by applying the equation
from the calibration curve (e.g., Fig. 2a). The response time should be deduced when
calculating $CH_4$ depth profiles and spatial distributions, in addition to the time lag caused by
pumping water samples by using an extended water hose.

### 4.3 Calibration, maintenance and mobility

The FaRAGE can be readily adopted for measuring other trace gases when coupled

with other portable gas analyzers. Due to differences in gas solubility (Duan and Sun, 2003;
Wiesenburg and Guinasso Jr, 1979), for each new gas, it would be necessary to establish the
relative equilibration efficiency and response time, following the approach we outlined here
for $CH_4$. Once set, a new calibration is only required when the tubing diameter or length is
changed (when the old one is filthy due to biofilm growth). This can be done by referring to a
number of known concentrations that covers a wide range (at least 5), e.g., taking water
samples from different water depth of the lake or a gradient from littoral to pelagic zones.





Once this full calibration is made, the calibration curve can be used for calculating the
subsequent measurements. A one-point reference measurement should be performed between
depth profiles or transects to check for apparent drifting. This can usually be done by taking
one surface water sample from a lake for manual headspace measurement. Care should be
taken when measuring in lakes with an anoxic hypolimnion where hydrogen sulfide is likely
to accumulate. The performance of Cavity-Ring-Down gas analyzers can be potential affected
by organics, ammonia, ethane, ethylene, or sulfur containing compounds (Kohl et al., 2019).
At these sites, it is always recommended to take additional samples and measure them with
traditional methods (e.g., with a Gas Chromatograph Analyzer).
The gas equilibrator should be carefully maintained. Replacement of parts is
recommended at a monthly basis provided the device is heavily in use. They include bubble
diffusor and the coiled gas-water mixing tube. In addition, to ensure the performance and
prevent biofilm formation the gas-water mixing and separation units should be cleaned after
use. Running with distilled or Milli-Q water would help to rinse the device and reduce the risk
of biofilm development in the inner tubes. The performance of peristaltic pumps should be
also regularly checked and the inner pump tubes need to be replaced to ensure a constant
water flow.
The combination of FaRAGE with the Picarro Gas Scouter provides the most mobility.
The system can be easily carried by one person and work in a small aluminum or inflatable
boat with a maximum capacity of three people is possible. The device can also work in bad
weather with additional measures based on protecting the gas analyzer from water damage by
rain or flooding.
**Code availability**
Not applicable.


**Data Availability**


An example calculation sheet (raw data of Fig. 2a) is provided as part of supporting


information for device calibration and for temperature and pressure correction when


calculating dissolved methane concentration. The full data sets associated with lab and field


tests are available upon request.


**Supplement link**


From Copernicus.


**Author contributions**


SBX and WW proposed the idea and built the first prototype. LL improved the


prototype and conducted lab and field tests. JW contributed to the field tests. AL contributed


to the derivation of equations; HPG advised the development of the modified prototype. LL


drafted the initial manuscript. All authors discussed the results and commented on the


manuscript.


**Competing interests**


The authors declare that they have no conflict of interest.


**Acknowledgements**


This work was financially supported by the National Natural Science Foundation of


China (grant No. 51979148) and Natural Science Foundation of Hubei Province, China


(2014CFB672). L.L., J.W. and H.P.G. were financially supported by the German Research


Foundation (DFG GR1540/21-1+2). The authors would like to thank Hannah Geisinger and


Truls Hveem Hansson for helping collecting field data.

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
