# Peer review of "A Fast-response automated gas equilibrator (FaRAGE) for continuous in situ"

_Hydrology and Earth System Sciences, 2020_

## Referee Comment (RC1) · Anonymous Referee #1 · 15 May 2020

General:

This study present a new gas equilibrator setup that makes it possible to perform dissolved CH4 measurements with short response time and at relatively low cost. The paper is very well written, the set-up is overall well described, and all relevant tests of the set-up are presented in a convincing manner. I think that this set-up will become widely used by people working on greenhouse gas dynamics in surface waters. I have no major concerns with this paper, just one major comment, and in addition a few minor comments that might help to further improve the paper.

Major comment: Given that gas analyzers were used that simultaneously measure both

CH4 and CO2, I really think that the authors should show the system's performance for CO2 as well. In L363, the authors write that have CO2 data but focus on CH4 for simplicity, but this choice makes life much less simple for all the researchers that want to measure both CH4 and CO2, and therefore need to do all the CO2 testing themselves. Showing the CO2 results would make this study much more useful and applicable for a much wider community, and certainly render more citations to this paper. At the very minimum, include the CO2 performance tests in the supplementary information, but I'd rather see that the CO2 data is fully integrated in the paper, including the title.

Minor comments:

Title: Include "carbon dioxide".

L13. Freshwater lakes and reservoirs are aquatic systems, so that's a repetitive formulation. Simplify.

L48-49. This sentence omits that dissolved CH4 concentration is very strongly a function of methanogenesis, this should be added.

L69. Not only phytoplankton, but also other microbial life forms. I suggest to reword to "biological".

Figure 1. The heart of the equilibrator is the gas-water mixing unit, and the gas-water separation unit. These should be illustrated much more clearly, as a technical drawing, such that people can build them themselves. The pictures in the SI don't really help very much.

L150-154. This text could go to the figure caption.

L155. Use full word "Laboratory"

L178. The 13 m tubing length is not mentioned in this list, but it's shown in the figures.

L193. Mention which these various methods were.

L204. From what I read, the depth from which peristaltic pumps can pump up water is physically limited to about 14 m. So how come you could pump water from 30 m depth?

L210. Was the effect of boat speed on equilibration tested? Depending on the type and placement of water intake, bubbles might start to form when driving too fast.

L229. Please give this correction equation.

L255. No details on statistical methods or tests are given, yet it says "significantly" here. Which test were performed, and what test statistics did they return?

L260. Please rephrase, "while extended response times" is unclear. Fig.2, panel d. Change the right y-axis colour to red (such as in panel c). Also, why is the red point for 13 m tubing length not connected to the other red points, and how come that its response time is so much longer for high-to-low than for low-to-high, and also so much longer than for the 8 m tubing?

L373. I would be more careful with this statement. You can state that the equilibrator was not negatively affected by high phytoplankton density, but you haven't tested suspended sediment, so it's not sure it would work in e.g. in a turbid river. You can't exclude that for sustained operation in a turbid system, a filter in the water intake might be required.

L383. Unclear what this sentence means, please rephrase.

L408. "potentially be" instead of "be potential".

Supporting information:

L32. Coupling instead of couple.

Fig.S1. The pictures of the syringes don't show much, and don't help those who want to build their own. Use better pictures and include a technical drawing.

L120. This is not a complete sentence.

Fig.S3. Please also show the corresponding depth profiles of $CH_4$ and $CO_2$ at this sampling occasion, such that the reader can judge in how far phytoplankton density might have affected measurements.

―――――――――――――――

---

## Referee Comment (RC2) · Anonymous Referee #2 · 20 May 2020

The information provided with this Technical Note is clear, well presented, and sufficiently useful for the scientific community. The idea of an equilibrator for measuring dissolved gas species in situ is certainly not new, but this work may deserve consideration cause it finds a clever way to make the measurement faster for dissolved methane. The set up takes advantage of the relatively new technology that led to small, fast responding gas spectrometers with quite big measuring cells and couples it to a small flow-through system to equilibrate/separate water and gas. The system is presented in a way that can be easily reproduced with affordable costs, hopefully promoting a long lasting need to resolve and map the spatial heterogeneity of dissolved methane in inland waters. It is indeed desirable for the measurement technique to achieve the

highest surface/time ratio.

I have two remarks to be considered and few suggestions 1)I was very interested in the CH4 profiles and - while impressed by the high vertical resolution they achieve – I understood the authors tested their system in different lakes, covering some "typical" methane concentration range. Indeed they show two profiles: one from meso-oligotrophic Lake Stechlin and one from eutrophic Lake Arend – I supposed they choose trophic state as a proxy for GHG content (Beaulieu et al., 2019). Thus I was confused seeing that Lake Arend, that they present as eutrophic, shows especially low methane, even lower than what they show for the meso-oligotrophic (Lake Stechlin). So what is the criterion behind this choice? Why picking an eutrophic lake that has even less methane than the meso-oligotrophic? Why not a typical eutrophic lake with methane building up below the oxycline during summer stratification? (They show a large literature on this issue in the Intro). For calibration the authors limited the upper range of methane to <2 micromoles L-1. While for high concentrations (microM to mM range) the authors suggest dilutions (line 357) but they don't deal with the problem in this paper. I think the range they show is fair enough, but they should clarify this "lower range test", directly relatable to surface waters but not to littoral methane rich or eutrophic lake bottom waters, AND change the sentence in the abstract "The FaRAGE is capable of continuously measuring dissolved CH4 concentrations in the nM-to-mM range" as it may be capable of that, but is not shown here. 2) One big step forward the authors claim is the "extraordinary fast response relative to all existing gas equilibration devices" (line 274). However, when their system is used with PICARRO G2132-1 + a desiccant to measure stable isotopes of gas species, it does not get that much faster than Hartmann 2018 "High Spatio-Temporal Dynamics of Methane Production and Emission in Oxic Surface Water" (line 107 of supplementary material). If I understand correctly then, what showed in table S4 is not entirely correct since their system response time when using PICARRO G2132-1 is obtained without any dryer. As far as I know, for using a PICARRO G2132-1 with a moisty flow a dryer is absolutely necessary. How humid is the water entering the analyzer? In case the authors

think that a dryer is needed, they should change the table S4 and the statement in line 274 " The comparison between up-to-date previous studies and this study (Table S4) demonstrated the extraordinary fast response relative to all existing gas equilibration devices. A 53 s response time was achieved when the FaRAGE was adapted to the Picarro G2132-i, which is significantly faster than others (171-6744 s)." Suggestions When describing the set up the authors often refer to a "bubble unit", which I suppose in the scheme (Fig.1) is called "gas-water mixing unit". Consider harmonize Line 363- I think it would be better to rephrase the reason why CO2 is not shown. "for simplicity" for them or for the reader? Maybe they can mention which non-simple problems do we meet when applying the system to CO2. As to line 366 it is possible for the authors to use their system at sea enhancing the liquid to HS ratio to achieve low concentrations. I would recommend to make sure that the scientific community that works on GHG air-water exchange in oceans gets interested too (add to abstract and line 97?). For different reasons from the ones highlighted here for inland waters (to name one the massive lack of ground data to calibrate satellite infers) this system could be applied to voluntary observing ship programs to map CO2 and CH4 surface concentrations. In case the authors find a major obstacle to this it would be good to mention - making a suggestion for adapting their system for oceanographic applications. Line 382- they mention how temperature should be corrected for the change along the hose- can give an example on how off can it get and does that mean to always measure temperature in situ at depth along with the profile? Line 400 replace "filthy" Line 410- wouldn't a scrubber serve for that? Would that slow down the system RT?

References Beaulieu, J.J., DelSontro, T. & Downing, J.A. Eutrophication will increase methane emissions from lakes and impoundments during the 21st century. Nat Commun 10, 1375 (2019). https://doi.org/10.1038/s41467-019-09100-5
* * *

---

## Author Comment (AC1) · 1 Jun 2020

Response to Anonymous Referee #1

Major comment: Given that gas analyzers were used that simultaneously measure both $CH_4$ and $CO_2$, I really think that the authors should show the system's performance for $CO_2$ as well. In L363, the authors write that have $CO_2$ data but focus on $CH_4$ for simplicity, but this choice makes life much less simple for all the researchers that want to measure both $CH_4$ and $CO_2$, and therefore need to do all the $CO_2$ testing themselves. Showing the $CO_2$ results would make this study much more useful and applicable for a much wider community, and certainly render more citations to this paper. At the very

minimum, include the CO2 performance tests in the supplementary information, but I'd rather see that the CO2 data is fully integrated in the paper, including the title.

Response: We totally agree with the reviewer that CO2 should be included to the manuscript. We will integrate CO2 throughout the manuscript and include CO2 into the title. Minor comments:

Title: Include "carbon dioxide".

Response: We will include CO2 in the revised manuscript.

L13. Freshwater lakes and reservoirs are aquatic systems, so that's a repetitive formulation. Simplify.

Response: We will change "freshwater lakes and reservoirs" to aquatic systems.

L48-49. This sentence omits that dissolved CH4 concentration is very strongly a function of methanogenesis, this should be added.

Response: We will add "In addition to formation processes that lead to CH4 accumulation" in the revised manuscript.

L69. Not only phytoplankton, but also other microbial life forms. I suggest to reword to "biological".

Response: We agree that many microorganisms might be involved and thus the word "biological" is more appropriate. We will make this change in the revised manuscript.

Figure 1. The heart of the equilibrator is the gas-water mixing unit, and the gas-water separation unit. These should be illustrated much more clearly, as a technical drawing, such that people can build them themselves. The pictures in the SI don't really help very much.

Response: We agree the drawing should be improved. We will prepare a technical drawing to replace Figure 1.

[Figure]

L150-154. This text could go to the figure caption.

Response: This text is already part of the figure caption.

L155. Use full word "Laboratory"

Response: We will change "Lab" to "Laboratory".

L178. The 13 m tubing length is not mentioned in this list, but it's shown in the figures.
Response: Thanks for pointing this out. We will add 13 m to the text.

L193. Mention which these various methods were.

Response: We will describe explicitly what the methods are in the text.

L204. From what I read, the depth from which peristaltic pumps can pump up water is physically limited to about 14 m. So how come you could pump water from 30 m depth?

Response: Pump head that the pump needs to overcome is related to vertical distance of the pump to water surface only and unrelated to the vertical position of water intake. Thus, often there is only < 0.5 m pump head when the FaRAGE is placed in a small boat.

L210. Was the effect of boat speed on equilibration tested? Depending on the type and placement of water intake, bubbles might start to form when driving too fast.

Response: We did not try speed higher than 10 km h-1. The driving speed should be chosen according to the spatial resolution that the users would like to have. In our case, 17 m spatial resolution (spatially averaged) was achieved at 5 km h-1 driving speed corresponding to 12 s response time (see line 322-324). Bubbles were not observed at 10 km h-1 speed when the water intake is mounted on the side wall of the boat, 0.5 m below water surface. Driving too fast is not recommended as it may harm the CTD probe.

L229. Please give this correction equation.

Response: The correction equations for CH4 and CO2 are shown in Fig. 2a and b, respectively. L255. No details on statistical methods or tests are given, yet it says "significantly" here. Which test were performed, and what test statistics did they return? Response: Thanks. Indeed, statistical tests were not performed. We will change this word to "substantially" and give mean $\pm$ standard derivation.

L260. Please rephrase, "while extended response times" is unclear. Fig.2, panel d. Change the right y-axis colour to red (such as in panel c). Also, why is the red point for 13 m tubing length not connected to the other red points, and how come that its response time is so much longer for high-to-low than for low-to-high, and also so much longer than for the 8 m tubing?

Response: Thanks. This is an incomplete sentence. We rephrase the sentence to "A 91.8% equilibration ratio can be achieved by extending the tube length to 13 m while extended response times are expected." We will change the right-handed y-axis color to red in panel d. The red point for 13 m tube length severely deviated from the well-fitted power function. The reason is partially mentioned in line 262-264. The sharp increase in response time of high-to-low is a result of increased resistance of the gas-water mixture flow. The instability started from 13 m tube length and became unacceptable when tube length is 18 m. We will add a few words to explain this a bit more.

L373. I would be more careful with this statement. You can state that the equilibrator was not negatively affected by high phytoplankton density, but you haven't tested suspended sediment, so it's not sure it would work in e.g. in a turbid river. You can't exclude that for sustained operation in a turbid system, a filter in the water intake might be required.

Response: Thanks. Indeed, so far the device has never been tested particularly in turbid rivers with suspended sediment particles. We will point this out explicitly and

suggest that a filtration unit for the water intake might be needed in turbid rivers.

L383. Unclear what this sentence means, please rephrase.

Response: Thanks. We will correct this in the text.

L408. "potentially be" instead of "be potential".

Response: Thanks. We will correct this in the text.

Supporting information: L32. Coupling instead of couple.

Response: We will correct this in the text.

Fig.S1. The pictures of the syringes don't show much, and don't help those who want to build their own. Use better pictures and include a technical drawing.

Response: We will improve these in the revised draft.

L120. This is not a complete sentence.

Response: Thanks. We will rephrase this sentence.

Fig.S3. Please also show the corresponding depth profiles of CH4 and CO2 at this sampling occasion, such that the reader can judge in how far phytoplankton density might have affected measurements.

Response: We will add depth profiles of CH4 and CO2 into Fig. S3.

―――――――――――――

---

## Author Comment (AC2) · 1 Jun 2020

Response to Anonymous Referee #2

I have two remarks to be considered and few suggestions 1) I was very interested in the CH4 profiles and - while impressed by the high vertical resolution they achieve – I understood the authors tested their system in different lakes, covering some "typical" methane concentration range. Indeed they show two profiles: one from mesooligotrophic Lake Stechlin and one from eutrophic Lake Arend – I supposed they choose trophic state as a proxy for GHG content (Beaulieu et al., 2019). Thus I was confused seeing that Lake Arend, that they present as eutrophic, shows especially low methane,

even lower than what they show for the meso-oligotrophic (Lake Stechlin). So what is the criterion behind this choice? Why picking an eutrophic lake that has even less methane than the meso-oligotrophic? Why not a typical eutrophic lake with methane building up below the oxycline during summer stratification? (They show a large literature on this issue in the Intro). For calibration the authors limited the upper range of methane to <2 micromoles L-1. While for high concentrations (microM to mM range) the authors suggest dilutions (line 357) but they don't deal with the problem in this paper. I think the range they show is fair enough, but they should clarify this "lower range test", directly relatable to surface waters but not to littoral methane rich or eutrophic lake bottom waters, AND change the sentence in the abstract "The FaRAGE is capable of continuously measuring dissolved CH4 concentrations in the nM-to-mM range" as it may be capable of that, but is not shown here.

Response: Thanks for the reviewer's very detailed comments. We have such CH4 profiles with a typical anoxic hypolimnion in summer where CH4 was enriched to sub milli-molar high concentration. We will add these data into the revised draft.

Also, we performed additional laboratory tests for high concentrations (e.g., 33 microM) and will add these to the new revision. The measured concentration is about 245 ppm with the FaRAGE. For the Gas Scouter 4301 we use, a linearity can only be guaranteed up to 500 ppm. Adjustment of water-gas mixing ratio is needed for mM CH4 concentrations. Therefore, we agree with the reviewer's suggestion. We will rephrase the measurement range-related statement accordingly.

2) One big step forward the authors claim is the "extraordinary fast response relative to all existing gas equilibration devices" (line 274). However, when their system is used with PICARRO G2132-1 + a desiccant to measure stable isotopes of gas species, it does not get that much faster than Hartmann 2018 "High Spatio-Temporal Dynamics of Methane Production and Emission in Oxic Surface Water" (line 107 of supplementary material). If I understand correctly then, what showed in table S4 is not entirely correct since their system response time when using PICARRO G2132-1 is obtained

without any dryer. As far as I know, for using a PICARRO G2132-1 with a moisty flow a dryer is absolutely necessary. How humid is the water entering the analyzer? In case the authors think that a dryer is needed, they should change the table S4 and the statement in line 274 "The comparison between up-to-date previous studies and this study (Table S4) demonstrated the extraordinary fast response relative to all existing gas equilibration devices. A 53 s response time was achieved when the FaRAGE was adapted to the Picarro G2132-i, which is significantly faster than others (171-6744 s)."

Response: Thanks for suggesting this. We did not have appropriate dryers for testing when the tests were performed. Therefore a dryer made from silicone beads was tested and 150 s extension in response time was observed (Line 108-109 in SI). But we are aware that this has been well tested in Webb et al (2016), in which they tested both Drierite and magnesium perchlorate (Mg(ClO4)2) as dryers. I reproduced their results below. They show both dryers have no effect on CH4 except 1.5 m time delay on CO2 was caused by using Drierite.

Yes, the water vapor content of gas sample flow is above 1% and should be dried before entering PICARRO G2132-i. We are currently using for these drying materials as suggested by Webb et al (2016) and they do a good job.

So the numbers in table S4 and the statement in line 274 are all valid. We understand it is not well clarified. We will clarify this in the revised draft.

Reference Webb, J. R., Maher, D. T., and Santos, I. R.: Automated, in situ measurements of dissolved CO2, CH4, and $\delta$13 C values using cavity enhanced laser absorption spectrometry: Comparing response times of air‐water equilibrators, Limnol. Oceanogr.: Methods, 14, 323-337, https://doi.org/10.1002/lom3.10092, 2016.

Suggestions When describing the set up the authors often refer to a "bubble unit", which I suppose in the scheme (Fig.1) is called "gas-water mixing unit". Consider harmonize.

Response: Thanks for pointing this out. We will check this carefully and make sure the

terms of parts are used consistently throughout the manuscript.

Line 363- I think it would be better to rephrase the reason why $CO_2$ is not shown. "for simplicity" for them or for the reader? Maybe they can mention which non-simple problems we meet when applying the system to $CO_2$.

Response: We will add $CO_2$ as well and incorporate throughout the full text.

As to line 366 it is possible for the authors to use their system at sea enhancing the liquid to HS ratio to achieve low concentrations.

Response: Thanks for suggesting this. We replaced one calibration point in Fig. 2, in which 5.5 nM dissolved $CH_4$ concentration can be well characterized (with the 500 mL min-1 to 1000 mL min-1 water-gas mixing ratio). Indeed, lower concentrations in the ocean system can be also well measured by increasing water-gas mixing ratio.

I would recommend to make sure that the scientific community that works on GHG air-water exchange in oceans gets interested too (add to abstract and line 97?). For different reasons from the ones highlighted here for inland waters (to name one the massive lack of ground data to calibrate satellite infers) this system could be applied to voluntary observing ship programs to map $CO_2$ and $CH_4$ surface concentrations. In case the authors find a major obstacle to this it would be good to mention – making a suggestion for adapting their system for oceanographic applications.

Response: Thanks for suggesting these. We agree that the FaRAGE can be a good method for studying GHGs from oceans. We will mention this explicitly in abstract and also in text. The potential use what the reviewer suggested is very interesting!

Line 382- they mention how temperature should be corrected for the change along the hose- can give an example on how off can it get and does that mean to always measure temperature in situ at depth along with the profile?

Response: We will include an example in SI to show that how different the water temperature (water flow inside the gas-water separation unit) can be from the in situ water

temperature. It's more important to monitor temperature of water flow inside the gas-water separation unit of the device. We recommend to install a temperature logger in the device if someone would like to rebuild the device but can only afford one thermometer.

Line 400 replace "filthy"

Response: Thanks. We will rephrase this sentence and replace "filthy".

Line 410- wouldn't a scrubber serve for that? Would that slow down the system RT?

Response: Thanks. Yes, a copper scrubber can help removing H2S gas from the gas samples (Malowany et al. 2015). According to Malowany et al. (2015), no time delay was observed when a copper scrubber was used. I also reproduced the figure they included in the publication. We will add this reference to the revised manuscript.

Reference Malowany, K., Stix, J., Van Pelt, A., and Lucic, G.: H2S interference on CO2 isotopic measurements using a Picarro G1101-i cavity ring-down spectrometer, Atmos. Meas. Tech., 8, 4075– 4082, https://doi.org/10.5194/amt-8-4075-2015, 2015.

References Beaulieu, J.J., DelSontro, T. & Downing, J.A. Eutrophication will increase methane emissions from lakes and impoundments during the 21st century. Nat Commun 10, 1375 (2019). https://doi.org/10.1038/s41467-019-09100-5

[Figure]

**Fig. 2.** Step experiments carried out to determine the effect of two desiccants, Drierite (**A**) and magnesium perchlorate ($Mg(ClO_4)_2$) (**B**), on the response times of $CO_2$ and $CH_4$ through a showerhead equilibrator. A time delay of 1.5 min was observed for $CO_2$ concentrations when Drierite was used as a desiccant. $Mg(ClO_4)_2$ had no effect on concentration response times and $CH_4$ remained unaffected by both desiccants.

**Fig. 1.** The effect of dryers on response time

[Figure]

**Figure 3.** Change in the $^{12}CO_2$ and $^{13}CO_2$ concentrations with addition of H$_2$S to the standard gas. **(a)** Plot showing the percentage change in CO$_2$ concentration between gas with H$_2$S and gas scrubbed of H$_2$S. There is a visible increase in the $^{12}CO_2$ concentration and a decrease in the $^{13}CO_2$ concentration with addition of H$_2$S. The percentage decrease for $^{13}CO_2$ is significantly greater than the percentage increase for $^{12}CO_2$. **(b)** Plot showing the 1000 ppm standard CO$_2$ gas with the addition of 3 mL of 100 ppm H$_2$S and the subsequent response after the H$_2$S was removed with the copper scrub. There is a small, yet visible, increase in the $^{13}CO_2$ concentration and decrease in the $^{12}CO_2$ concentration when H$_2$S is removed.

**Fig. 2.** The effect of copper scrubber on response time

---

## Author Response (AR1)

**Response to Anonymous Referee #1**

**Major comment**: Given that gas analyzers were used that simultaneously measure both CH4 and CO2, I really think that the authors should show the system's performance for CO2 as well. In L363, the authors write that have CO2 data but focus on CH4 for simplicity, but this choice makes life much less simple for all the researchers that want to measure both CH4 and CO2, and therefore need to do all the CO2 testing themselves. Showing the CO2 results would make this study much more useful and applicable for a much wider community, and certainly render more citations to this paper. At the very minimum, include the CO2 performance tests in the supplementary information, but I'd rather see that the CO2 data is fully integrated in the paper, including the title.

**Response**: We totally agree with the reviewer that CO2 should be included in the manuscript. We integrated CO2 results in the manuscript and mention CO2 in the title as well. Please see the manuscript.

**Minor comments:**
Title: Include "carbon dioxide".
**Response**: We included CO2 in the revised manuscript.

L13. Freshwater lakes and reservoirs are aquatic systems, so that's a repetitive formulation. Simplify.
**Response**: We deleted "in freshwater lakes and reservoirs" in Line 13. Please see line 14-15 in the revision.

L48-49. This sentence omits that dissolved CH4 concentration is very strongly a function of methanogenesis, this should be added.
**Response**: We added "In addition to formation processes that lead to CH4 accumulation" in the revised manuscript. Please see line 61-62 in the revision.

L69. Not only phytoplankton, but also other microbial life forms. I suggest to reword to "biological".

**Response**: We agree that many microorganisms might be involved and thus the word "biological" is more appropriate. We made this change in the revised manuscript (Line 91 in the marked revision).

Figure 1. The heart of the equilibrator is the gas-water mixing unit, and the gas-water separation unit. These should be illustrated much more clearly, as a technical drawing, such that people can build them themselves. The pictures in the SI don't really help very much.

**Response**: We have replaced Figure S1 with a technical drawing in which the gas-water mixing unit and the gas-water separation unit were both described in detail.

L150-154. This text could go to the figure caption.

**Response**: This text has been removed from the main text and now is part of the figure caption. Please see line 172-176 in the revision.

L155. Use full word "Laboratory"

**Response**: We changed "Lab" to "Laboratory". Please see line 177.

L178. The 13 m tubing length is not mentioned in this list, but it's shown in the figures.

**Response**: Thanks for pointing this out. We added 13 m to the text. Please see line 204.

L193. Mention which these various methods were.

**Response**: We described explicitly what the methods are in the text. Please see line 218-219.

L204. From what I read, the depth from which peristaltic pumps can pump up water is physically limited to about 14 m. So how come you could pump water from 30 m depth?

**Response**: Pump head that the pump needs to overcome is related to vertical distance of the pump to water surface only and unrelated to the vertical position of water intake below the water surface. Thus, often there is only < 0.5 m pump head when the FaRAGE is placed in a small boat.

L210. Was the effect of boat speed on equilibration tested? Depending on the type and placement of water intake, bubbles might start to form when driving too fast.

**Response**: We did not try higher boat speed than 10 km h-1. The driving speed should be chosen according to the spatial resolution that the users would like to have. In our case, 17 m spatial resolution (spatially averaged) was achieved at 5 km h-1 driving speed corresponding to 12 s response time (see line 377-378). Bubbles were not observed at 10 km h-1 speed when the water intake is mounted on the side wall of the boat, 0.5 m below the water surface. Driving too fast is not recommended as it may harm the CTD probe as well.

L229. Please give this correction equation.

**Response**: The correction equations for CH4 and CO2 are shown in Fig. 2a and b, respectively.

L255. No details on statistical methods or tests are given, yet it says "significantly" here. Which test were performed, and what test statistics did they return?

**Response**: Thanks. Indeed, statistical tests were not performed. We changed this word to "substantially" and gave mean ± standard deviation. Please see line 289-290.

L260. Please rephrase, "while extended response times" is unclear. Fig.2, panel d. Change the right y-axis colour to red (such as in panel c). Also, why is the red point for 13 m tubing length not connected to the other red points, and how come that its response time is so much longer for high-to-low than for low-to-high, and also so much longer than for the 8 m tubing?

**Response**: Thanks. This is an incomplete sentence. We rephrased the sentence to "A 91.8% equilibration ratio can be achieved by extending the tube length to 13 m while extended response times are expected."

We have changed the right-handed y-axis color to red in panel d. Please see the new Fig. 3c-d. The red point for the 13 m tube length severely deviated from the well-fitted power function. The sharp increase in response time of high-to-low is a result of increased resistance of the gas-water mixture flow. The instability (abnormal sharp increase in response time) started from 13 m tube length and became unacceptable when the tube length is 18 m. Please see our explanations in line 294-302.

L373. I would be more careful with this statement. You can state that the equilibrator was not negatively affected by high phytoplankton density, but you haven't tested suspended sediment, so it's not sure it would work in e.g. in a turbid river. You can't exclude that for sustained operation in a turbid system, a filter in the water intake might be required.

**Response**: Thanks. Indeed, so far the device has never been tested particularly in turbid rivers with suspended sediment particles. We confined this statement to lakes without high sediment loads and also pointed out explicitly that a filtration unit for the water intake might be needed in turbid rivers. Please see line 427-433.

L383. Unclear what this sentence means, please rephrase.

**Response**: Thanks. They are indeed misleading and unnecessary, therefore, we have removed them from the manuscript.

L408. "potentially be" instead of "be potential".

**Response**: We corrected this in the text. Please see line 466.

**Supporting information:**

L32. Coupling instead of couple.

**Response**: We corrected this in the text (line 33).

Fig.S1. The pictures of the syringes don't show much, and don't help those who want to build their own. Use better pictures and include a technical drawing.

**Response**: We improved these by replacing them with technical drawings.

L120. This is not a complete sentence.

**Response**: Thanks. We rephrased this sentence. Please see line 127-128.

Fig.S3. Please also show the corresponding depth profiles of CH4 and CO2 at this sampling occasion, such that the reader can judge in how far phytoplankton density might have affected measurements.

**Response**: We added depth profiles of CH4 and CO2 to Fig. S3.

**Response to Anonymous Referee #2**

**I have two remarks to be considered and few suggestions**

1) I was very interested in the CH4 profiles and - while impressed by the high vertical resolution they achieve – I understood the authors tested their system in different lakes, covering some "typical" methane concentration range. Indeed they show two profiles: one from mesooligotrophic Lake Stechlin and one from eutrophic Lake Arend – I supposed they choose trophic state as a proxy for GHG content (Beaulieu et al., 2019). Thus I was confused seeing that Lake Arend, that they present as eutrophic, shows especially low methane, even lower than what they show for the meso-oligotrophic (Lake Stechlin). So what is the criterion behind this choice? Why picking an eutrophic lake that has even less methane than the meso-oligotrophic? Why not a typical eutrophic lake with methane building up below the oxycline during summer stratification? (They show a large literature on this issue in the Intro). For calibration the authors limited the upper range of methane to <2 micromoles L-1. While for high concentrations (microM to mM range) the authors suggest dilutions (line 357) but they don't deal with the problem in this paper. I think the range they show is fair enough, but they should clarify this "lower range test", directly relatable to surface waters but not to littoral methane rich or eutrophic lake bottom waters, AND change the sentence in the abstract "The FaRAGE is capable of continuously measuring dissolved CH4 concentrations in the nM-to-mM range" as it may be capable of that, but is not shown here.

**Response**:

Thanks for the reviewer's very detailed comments. We have such CH4 profiles with a typical anoxic hypolimnion in summer where CH4 was enriched to sub milli-molar high concentration. We have added these data into the revised manuscript. Please see Fig. 4c-d.

We further added additional laboratory tests for high CH4 concentration (e.g., 33 μM). The measured concentration is about 245 ppm with the FaRAGE. For the Gas Scouter 4301 we use, linearity can only be guaranteed up to 500 ppm. Adjustment of water-gas mixing ratio is needed for CH4 concentrations in the mM range. Therefore, we agree with the reviewer's suggestion. We rephrased the measurement range-related statement accordingly.

2) One big step forward the authors claim is the "extraordinary fast response relative to all existing gas equilibration devices" (line 274). However, when their system is used with PICARRO G2132-1 + a desiccant to measure stable isotopes of gas species, it does not get that much faster than Hartmann 2018 "High Spatio-Temporal Dynamics of Methane Production and Emission in Oxic Surface Water" (line 107 of supplementary material). If I understand correctly then, what showed in table S4 is not entirely correct since their system response time when using PICARRO G2132-1 is obtained without any dryer. As far as I know, for using a PICARRO G2132-1 with a moisty flow a dryer is absolutely necessary. How humid is the water entering the analyzer? In case the authors think that a dryer is needed, they should change the table S4 and the statement in line 274 "The comparison between up-to-date previous studies and this study (Table S4) demonstrated the extraordinary fast response relative to all existing gas equilibration devices. A 53 s response time was achieved when the FaRAGE was adapted to the Picarro G2132-i, which is significantly faster than others (171-6744 s)."

**Response**:

Thanks for suggesting this. We did not have appropriate dryers for testing when the tests were performed. Therefore a dryer made from silicone beads was tested and a 150 s extension in response time was observed. But we are aware that this has been well tested in Webb et al (2016), in which they tested both Drierite and magnesium perchlorate (Mg(ClO4)2) as dryers. We reproduced their results below. They show both types of dryer have no effect on CH4 and CO2, except for a 1.5 min time delay on CO2 was caused by using Drierite.

Yes, the water vapor content of the gas sample flow is above 1% and should be dried before entering the PICARRO G2132-i. We are currently using the drying materials as suggested by Webb et al (2016) and they work quite well.

So the numbers in table S4 and the statement in line 274 are all valid. We understand it is not well clarified. We clarified this in the revised version of the manuscript. Please see the revision in line 113-117 in the supporting information.

**Reference**

Webb, J. R., Maher, D. T., and Santos, I. R.: Automated, in situ measurements of dissolved $CO_2$, $CH_4$, and $\delta_{13}$ C values using cavity enhanced laser absorption spectrometry: Comparing response times of air-water equilibrators, Limnol. Oceanogr.: Methods, 14, 323-337, https://doi.org/10.1002/lom3.10092, 2016.

[Figure]

**Fig. 2.** Step experiments carried out to determine the effect of two desiccants, Drierite (**A**) and magnesium perchlorate (Mg(ClO₄)₂) (**B**), on the response times of $CO_2$ and $CH_4$ through a showerhead equilibrator. A time delay of 1.5 min was observed for $CO_2$ concentrations when Drierite was used as a desiccant. Mg(ClO₄)₂ had no effect on concentration response times and $CH_4$ remained unaffected by both desiccants.

**Suggestions**

When describing the set up the authors often refer to a "bubble unit", which I suppose in the scheme (Fig.1) is called "gas-water mixing unit". Consider harmonize.

**Response**:

Thanks for pointing this out. We checked this carefully and now the terms are used consistently throughout the manuscript.

Line 363- I think it would be better to rephrase the reason why CO2 is not shown. "for simplicity" for them or for the reader? Maybe they can mention which non-simple problems we meet when applying the system to CO2.

**Response**:

We have added CO2 as well and incorporated it throughout the full manuscript.

As to line 366 it is possible for the authors to use their system at sea enhancing the liquid to HS ratio to achieve low concentrations.

**Response**:

Thanks for suggesting this. We replaced one calibration point in Fig. 2, in which 5.5 nM dissolved CH4 concentration can be well characterized (with the 500 mL min-1 to 1000 mL min-1 water-gas mixing ratio). Indeed, lower concentrations in the ocean system can be also well measured by increasing water-gas mixing ratio.

I would recommend to make sure that the scientific community that works on GHG air-water exchange in oceans gets interested too (add to abstract and line 97?). For different reasons from the ones highlighted here for inland waters (to name one the massive lack of ground data to calibrate satellite infers) this system could be applied to voluntary observing ship programs to map CO2 and CH4 surface concentrations. In case the authors find a major obstacle to this it would be good to mention – making a suggestion for adapting their system for oceanographic applications.

**Response**:

Thanks for suggesting these. We agree that the FaRAGE can be a good method for studying GHGs from oceans. We added a sentence in abstract (line 35-36) and also in text (line 117-118).

Line 382- they mention how temperature should be corrected for the change along the hose- can give an example on how off can it get and does that mean to always measure temperature in situ at depth along with the profile?

**Response**:

We included an example in SI (Fig. S4) to show how much the water temperature (water flow inside the gas-water separation unit) can differ from the in situ water temperature. It's more important to monitor temperature of water flow inside the gas-water separation unit of the device. We recommend to installing a temperature logger in the device if someone would like to rebuild the device, but can only afford one thermometer.

Line 400 replace "filthy"

**Response**:

Thanks. We rephrased this sentence (see line 458).

Line 410- wouldn't a scrubber serve for that? Would that slow down the system RT?

**Response**:

Thanks. Yes, a copper scrubber can help to remove H2S from the gas samples (Malowany et al. 2015). According to Malowany et al. (2015), no time delay was observed when a copper scrubber was used. We also reproduced the figure they included in the publication. We added this reference to the revised manuscript (line 469-471).

**Reference**

Malowany, K., Stix, J., Van Pelt, A., and Lucic, G.: H2S interference on CO2 isotopic measurements using a Picarro G1101-i cavity ring-down spectrometer, Atmos. Meas. Tech., 8, 4075– 4082, https://doi.org/10.5194/amt-8-4075-2015, 2015.

[Figure]

**Figure 3.** Change in the $^{12}CO_2$ and $^{13}CO_2$ concentrations with addition of $H_2S$ to the standard gas. **(a)** Plot showing the percentage change in $CO_2$ concentration between gas with $H_2S$ and gas scrubbed of $H_2S$. There is a visible increase in the $^{12}CO_2$ concentration and a decrease in the $^{13}CO_2$ concentration with addition of $H_2S$. The percentage decrease for $^{13}CO_2$ is significantly greater than the percentage increase for $^{12}CO_2$. **(b)** Plot showing the 1000 ppm standard $CO_2$ gas with the addition of 3 mL of 100 ppm $H_2S$ and the subsequent response after the $H_2S$ was removed with the copper scrub. There is a small, yet visible, increase in the $^{13}CO_2$ concentration and decrease in the $^{12}CO_2$ concentration when $H_2S$ is removed.

*Corresponding authors

Emails: liu.liu@igb-berlin.de; hgrossart@igb-berlin.de

**Abstract**

Biogenic greenhouse gas  emissions, e.g. of methane (CH$_4$) and carbon dioxide (CO$_2$) from inland waters contribute substantially to global warming. In aquatic systems, dissolved  greenhouse gases are highly heterogeneous both in space and time. To better understand the biological and physical processes that affect sources and sinks of both CH$_4$ and CO$_2$ , their dissolved concentrations need to be measured with  high spatial and temporal resolution. To achieve this goal, we developed the **Fast-Response Automated Gas Equilibrator (FaRAGE)** for real- time *in situ* measurement of dissolved CH$_4$ and CO$_2$ concentrations at the water surface and in the water column. FaRAGE can achieve an exceptionally short response time (t$_{95\%}$ = 12 s when including the response time of the gas analyzer) while retaining an equilibration ratio of 62.6%

and a measurement accuracy of 0.5% for CH$_4$. A similar performance was observed for dissolved $CO_2$ ($t_{95\%}$ = 10 s, equilibration ration 67.1%). An equilibration ratio, as high as 91.8%, can be reached at the cost of a slightly increased response time (16 s). The FaRAGE is capable of continuously measuring dissolved $CO_2$ and $CH_4$ concentrations in the nM-to-sub mM ($10^{-9}$ -

$10^{-3}$ mol $L^{-1}$) range with a detection limit of sub-nM ($10^{-10}$ mol $L^{-1}$), when couplinged with a cavity ring-down greenhouse gas analyzer (Picarro GasScouter). It enables the possibility of mapping dissolved $CH_4$ concentration in a "quasi" three-dimensional manner in lakes.

Additional tests demonstrated a sSimilarly good performance of FaRAGEthe equilibrator could be demonstrated for measuring dissolved $CO_2$. FaRAGE enablesallows 
[revised manuscript text omitted]

**S1. Details of parts, gas analyzers and costs**

To make the FaRAGE field deployable, parts were tightly packed into an aluminum box with a built-in power supply. The electric parts were separated from other parts containing water in the box by using a plastic board. Ports were well labelled on the right-handed side so that even somebody new to the system can work with it. To help interested readers rebuild the device, the two key components (gas-water mixing unit and gas-water separation unit) were shown in the detailed technical drawings (Fig. S1). The suppliers and costs for these parts were listed in Table S1. A total of 3,560 € was calculated for building the complete device excluding the costs for the power supply. As the expensive RBR temperature logger is not a necessity since we happen to have it in storage, a cheaper temperature logger can always be used. For example, a fast HOBO temperature logger (HOBO U12 with a Temperature probe TMC1-HD) is available for < 200 €. The total cost can be cut down significantly to < 3,000 €.

The FaRAGE is capable of coupling with different greenhouse gas analyzers, depending on the research question and instrument availability. Three most widely used field-deployable gas analyzers were compared in Table S2 to provide a reference for readers when choosing a gas analyzer. They are GasScouter G4301 (Picarro, USA), Ultraportable Greenhouse Gas Analyzer (Model 915-0011, LosGatos Research, USA) and Picarro G2132-i isotope analyzer (Picarro, USA). We noticed Picarro 2201-i has been more often used, but our Picarro G2132-i is an equivalent instrument except that the module for isotopic $CO_2$ is not installed. The former two instruments measure $CH_4$, $CO_2$ and $H_2O$ and the last one additionally measures stable isotopic $CH_4$. As shown in Table 2, clearly GasScouter G4301 is most suitable for field measurement of dissolved $CH_4$ concentrations due to its extremely high mobility. The built-in battery pack can support 8 h continuous measurements and the ability to amount GPS antenna offers the advantage in doing spatially-resolved measurements. The Picarro G2132-i isotope analyzer is most immobile because of it is heavy and relative high power consumption in addition to its particularly long time to warm up (30 min). However, Picarro G2132-i measures stable isotopic CH₄, while the other two instruments cannot. Care must be taken and a proper boat with stable power supply is needed in order to use Picarro G2132-i as a coupling unit for the FaRAGE.

[Figure]

[Figure]

**Fig. S1**  Technical drawings of FaRAGE key components.

(a) Gas-water mixing unit and (b) gas-water separation unit. Note: ID and OD are the abbreviations of inner diameter and outside diameter, respectively.

**Table S1.** List of materials for parts of the FaRAGE prototype. Details on dimensions, model, producer/supplier and cost are provided.

| Items | Dimensions | Model specifications | Producer/Supplier | Quantity | Cost |
|---|---|---|---|---|---|
| Diving tank | 10 L | Pressure up to 230 bar | Atlantis Berlin | 1 | 199 € |
| Pressure regulator | | 200 bar / 0 - 10 bar, HERCULES CK1401 | Gase Dopp | 1 | 59.98 € |
| Mass flow controller (for air) | | SIERRA Model C50L SMART-TRAK | SCHWING Verfahrenstechnik GmbH | 1 | 995 € |
| Flow meter (for water) | | 0.082-0.82 L min-1, 1355GAF3CBXN1AAA | Brooks Instrument GmbH | 1 | 943.91 € |
| Peristaltic pump | 9 x 11 x 16 cm | 0-500 mL min-1, 24V/1A DC power | Purchased from Taobao, China | 2 | 200 € |
| Temperature logger | | Precision 0.001 °C, maximum 6 Hz measurement frequency, TR-1050 | RBR, Canada | 1 | 1,000 € |
| Tygon tube | 3.2/6.4 mm in./out. Ø | Saint-Gobain Schlauch Tygon S3 E-3603 2.5bar | RS Components GmbH | 15 m | 68.78 € |
| Plastic syringe for mixing unit | 5 mL | Cut to 3 mL, sealed with a rubber stopper | BD plastipak | 1 | 1 € |
| Plastic syringe separation unit | 30 mL | Sealed with a rubber stopper | BD plastipak | 1 | 1 € |
| Plastic syringe for desiccant | 50 mL | Filled with silicone beads, sealed with a rubber stopper | BD plastipak | 1 | 1 € |
| Rain pipe | | | Toom | 1 | 10 € |
| Bubble diffusor | 12 mm Ø, 16 mm length | Pawfly 0.6 Inch Air Stone, UL266 | Ebay | 1 | 1 € |
| Teflon membrane filter | 25 mm Ø | PTFE 0.2 μm | Lab Logistics Group GmbH | 2 | 2 € |
| Tube connector | for 3.2-4.2 mm | LL male, barbed hose connection:  PP, 10 pcs/pack \| 2-1882 | neoLab Migge GmbH | 10 | 12 € |
| Aluminium box | 38.3 x 57 x 37.5 cm | 65 L, Stier aluminium box | Amazon | 1 | 64.95 € |
| Total | | | | | 3,560 € |

**Table S2.** Summary of technical details for the three greenhouse gas analyzers tested in this study.

| Analyzer | Gases | Gas flow rate | Cavity pressure | Measurement frequency | Concentration range | Precision | Response time | Dimensions | Weight | Power consumption | GPS Kit | Mobility |
|---|---|---|---|---|---|---|---|---|---|---|---|---|
| GasScouter G4301 | $CH_4$ $CO_2$ $H_2O$ | 1 L min$^{-1}$ | > 700 Torr | 1 Hz | $CH_4$: 0-800 ppm $CO_2$: 0-3% $H_2O$: < 3% | $CH_4$: 3 ppb $CO_2$: 0.4 ppm | 5 s | 35.6 × 17.7 × 46.4 cm | 10.4 kg | 25 W, built-in Li-ion battery | Yes | Very high |
| Ultraportable Greenhouse Gas Analyzer 915–0011 | $CH_4$ $CO_2$ $H_2O$ | 0.5 L min$^{-1}$ | 140 Torr | 1 Hz | $CH_4$: 0.01-100 ppm $CO_2$: 1-2% $H_2O$: < 7% | $CH_4$: 2 ppb $CO_2$: 0.6 ppm | ~10 s | 17.8 x 47 × 35.6 cm | 17 kg | 70 W, on battery/AC power | No | High |
| Picarro G2132-i | $CH_4$ $\delta^{13}C$-$CH_4$ $CO_2$ $H_2O$ | 25 mL min$^{-1}$ | 148 Torr | 0.5 Hz | $CH_4$: 1.8-10 ppm high-performance mode; 10-1000 high-range mode $CO_2$: 200 - 2000 ppm guaranteed range $H_2O$: <2.4 % guaranteed range | $CH_4$: 5 ppb + 0.05 % of reading (12C); 1 ppb + 0.05 % of reading (13C) $CO_2$: 1 ppm + 0.25 % of reading (12C) | ~30 s | 43.2 x 17.8 x 44.6 cm | 27.4 kg | 205 W, AC power | No | Fair |

Note: 1) GasScouter G4301 does not use a vacuum pump to maintain a stable cavity pressure and the gas flow rate should be stable but slightly above/below the recommended value.

2) All gas analyzers are sensitive to liquid-phase water, therefore a hydrophobic filter is normally placed before the gas intake to protect instrument from being flooded.

3) According to Picarro, interference can occur for concentrations of $H_2O$ and $CO_2$ well above normal ambient levels, as well as other organics, ammonia, ethane, ethylene, or sulfur containing compounds.

**S2. Re-evaluation of response time of gas analyzers**

While response time for each gas analyzer has been provided by its manufacturer (Table S2), a large difference was found when they were re-evaluated (Fig. S2). Picarro GasScouter has the fastest response to concentration increase, in comparison to four-fold and eight-fold slower response for portable Los Gatos and Picarro G2132-i, respectively. All three gas analyzers were seen longer response time when concentration changed from high to low. The Picarro GasScouter still has the best performance compared to the other two.

[Figure]

**Fig. S2** Response times of gas analyzers. Triplicated measurements were performed. Low-to- high and high-to-low concentration changes were investigated. The response time was determined by taking the time when 95% of final concentration was reached. For $\delta^{13}$C-CH$_4$,

30 s moving average data was used.

**S3. Theoretical background**

With the present design of the **Fast-Response Automated Gas Equilibrator (FaRAGE)**, a continuous dynamic gas-water mixing occurs and the carrier gas is partially equilibrating with the CH$_4$ dissolved in water sample. The gas composition reaching the gas analyzer depends on equilibration time and flow rates. The equilibration between the carrier gas and the water sample during flowing through the FaRAGE depends on the concentration difference between the gas stream ($C$ in µmol L$^{-1}$) and the dissolved (aqueous) concentration in the sample water ($C_a$):

$$\frac{dC}{dt} = k \times (\frac{1}{HRT} C_a - C) \qquad (1)$$

Where $H$ is the temperature-dependent Henry constant (mol L$^{-1}$ atm$^{-1}$), $R$ the universal gas constant (8.31 J mol$^{-1}$ K$^{-1}$), $T$ is temperature (K) and $k$ (s$^{-1}$) is an exchange coefficient. The equilibrium gaseous concentration $C_{eq} = \frac{1}{HRT} C_a$ corresponds to the headspace concentration of a fully equilibrated water sample. $k$ is expected to depend on the relative flow rates of gas and water as well as on the flow regime and mixing of both phases in the FaRAGE. For an initial concentration of CH$_4$ in the carrier gas $C_{ini}$, the time-dependent concentration during the passage through the equilibrator is:

$$C(t) = \left(C_{ini} - C_{eq}\right)e^{-kt} + C_{eq} \qquad (2)$$

After a device-specific partial equilibration time $t_e$, the CH$_4$ concentration in the carrier gas has changed to $C_{pe}$, which is measured by the gas analyzer

$$C_{pe} = C(t_e) = K\left(C_{ini} - C_{eq}\right) + C_{eq} \tag{3}$$

With $K = e^{-kt_e}$ being a device-specific coefficient, which can be obtained by calibrating

the FaRAGE with at least one water sample of known dissolved concentration ($C_{eq}$) through:

$$K = \frac{C_{pe} - C_{eq}}{C_{ini} - C_{eq}} \tag{4}$$

The equilibrium headspace concentration of $CH_4$ in the water sample and the

corresponding dissolved concentration can be estimated from the initial and final carrier gas

concentration as:

$$C_{eq} = \frac{1}{HRT} C_a = (KC_{ini} - C_{pe})/(K - 1) \tag{5}$$

For a high flow rate of the carrier gas, the response time of the system to changing

dissolved concentrations at the sample intake is predominantly determined by the gas venting

rate, i.e. by the total volume of carrier gas that is in contact with the water sample, divided by

the volumetric gas flow rate (cf. level two model of Johnson (1999)), as well as by the

response time of the gas analyzer.

**Table S3.** Response times when adapting to different gas analyzers. Tests were performed

with a water/gas mixing ratio of 0.5. Triplicates were made and mean values are shown here.

| Gas analyzer | Treatment | $t_{95\%}$ response time (s) | | |
|---|---|---|---|---|
| | | $CH_4$ | $CO_2$ | $^{13}\delta C\text{-}CH4$ |
| Gas Scouter G4301 | Low-to-high | 13 | 6 | - |
| | High-to-low | 13 | 6 | - |
| Ultraportable Greenhouse Gas Analyzer 915-0011 | Low-to-high | 34 | 32.3 | - |
| | High-to-low | 37 | 30 | - |
| Picarro G2132-i | Low-to-high | 53 | 53 | 53 |
| | High-to-low | 65.3 | 60.7 | 65.3 |

Note: Response time for Picarro G2132-i was determined without using a desiccant. A desiccant should be used to keep the moisture content in gas samples < 1%. Drierite and magnesium perchlorate $(Mg(ClO_4)_2)$ are recommended for such a purpose due to their high performance. It was shown by Webb et al. (2016) that both types of dryer had no effect on $CH_4$ and $CO_2$, except for a 1.5 min delay in response time for $CO_2$ when using Drierite.

**Table S4.** Comparison of response times for simultaneous measurement of dissolved $CH_4$ and $\delta^{13}C$-$CH_4$ in water from previous studies using different devices (after Webb et al., 2016, Hartman et al., 2018) compared with response times in this study. Response time was unified here to $t_{95\%}$ to allow for meaningful comparison. The $t_{95\%}$ values were taken from literature by applying $t_{95\%} = 3\tau$ and the mean were used.

| Device | $t_{95\%}$ response time (s) | Study |
|---|---|---|
| Weiss-type (small) | 6744 | Li et al. (2015) |
| General oceanics | 6123 | Webb et al. (2016) |
| Shower head | 4971 | Webb et al. (2016) |
| Weiss-type (large) | 3600 | Rhee et al. (2009) |
| Marble | 2679 | Webb et al. (2016) |
| Bubble-type | 2034 | Gülzow et al. (2011) |
| Liqui-Cel (medium) | 1251 | Webb et al. (2016) |
| Liqui-Cel (small) | 531 | Webb et al. (2016) |
| Liqui-Cel (large) | 351 | Webb et al. (2016) |
| Liqui-Cel (small) in vacuum mode | 171 | Hartmann et al. (2018) |
| Combined Weiss-type with bubble-type | 53 | This study |

**S4. The depth profiles of phytoplankton biomass at Lake Arend and Lake Stechlin**

As in most freshwater lakes phytoplankton is a large component of suspended solids in water column, the effect of phytoplankton biomass on the performance of the gas equilibrator was evaluated. Fig. S3 shows  the presence of  high phytoplankton biomass (represented by Chl-a) within the surface 20 m water depth in the both study lakes.

[Figure]

**Fig. S3** Depth profiles of Chlorophyll-a (Chl-a) at Lake Arend and Lake Stechlin on June 17

and July 23, 2019 with (b)-(c) dissolved $CH_4$ and $CO_2$ profiles. The profiles were measured using a BBE FluoroProbe (Moldaenke, Germany) simultaneously with dissolved gas profiles.

[Figure]

**Fig. S4** An example of altered depth profile of water temperature at Lake Stechlin in autumn

2019. (a) Comparison of in situ water temperature (red line) with water temperature measured in the FaRAGE (black line). (b)  The difference between the two temperature measurements (In FaRAGE - In situ).